The expression of metastasis associated protein 2 in normal development and cancers: mechanism and clinical significance

Liu Xujun 1
Jiang Yaping 1
Hou Yanfeng 1
Li Xiaoning 1
Li Haixia bdyylhx@126.com 1
Si Wenzhe wenzhesi@bjmu.edu.cn 2
1 Department of Laboratory Medicine, Peking University First Hospital , Beijing , China
2 Department of Laboratory Medicine, Peking University Third Hospital , Beijing , China
Mitsouras Katherine
Electronic publication date: 2025 Oct 24
Publication date: 2025
Volume: 13
Electronic Location ID: e20107
Received 2025 Feb 11; Accepted 2025 Aug 28
Copyright: ©2025 Liu et al.
Copyright year: 2025
Copyright holder: Liu et al.
License: This is an open access article distributed under the terms of the Creative Commons Attribution License, which permits unrestricted use, distribution, reproduction and adaptation in any medium and for any purpose provided that it is properly attributed. For attribution, the original author(s), title, publication source (PeerJ) and either DOI or URL of the article must be cited.
License URL: https://creativecommons.org/licenses/by/4.0/

Keywords: MTA2, Development, Cancer progression, Mechanism, Clinical significance

Funding: Beijing Nova Program 20220484090 20230484442 The National Natural Science Foundation of China 82303063 82072369 Beijing Natural Science Foundation 7232206 Clinical Medicine Plus X-Young Scholars Project of Peking University This work was supported by Beijing Nova Program (20220484090, 20230484442 to Wenzhe Si), the National Natural Science Foundation of China (82303063 to Xujun Liu, to 82072369 to Haixia Li) and Beijing Natural Science Foundation (7232206 to Wenzhe Si). Clinical Medicine Plus X-Young Scholars Project of Peking University. The funders had no role in study design, data collection and analysis, decision to publish, or preparation of the manuscript.

==============================
Metastasis-associated protein 2 (MTA2), a master transcriptional regulator, through multiple target genes and interacting proteins, has been demonstrated to play a vital role in the regulation of proliferation, replication, apoptosis, autophagy, DNA damage repair, preimplantation, embryonic development and immune cell differentiation. Despite extensive research, the physiological role and pathogenic mechanisms of MTA2 remain poorly understood. Here, we mainly review in the current research the status of MTA2 and its implications in normal development and various tumor biology. Accumulating evidence suggests that MTA2 is frequently amplify in several types of cancers, closely associates with tumor cells migration and invasion, relates to the malignant characteristics and poor prognosis, which therefore has been considered as playing tumor oncogenic roles. Substantial evidence indicates that MTA2 functions by modulating downstream targets including cell growth, invasion as well as angiogenesis related genes. Confusingly, the proliferation effect of MTA2 remains elusive and even conflicting in the development of several solid tumors. Furthermore, we discuss the upstream regulation of MTA2 by transcription factors, microRNAs and lncRNAs in specific physiology and pathology conditions, which results in the abnormal MTA2 expression in various aspects of cancer. In this context, we summarize linked function of MTA2 directly to oncogenesis and might provide a significant avenue for the treatment of diseases. We hope that this review will help tumor molecular biologists further understand the molecular mechanism of MTA2 in normal development and cancer.

Introduction

Tumor metastasis is the major cause of cancer patients mortality and one of the important hallmarks of cancer (Hanahan & Weinberg, 2011; Makitie et al., 2019). Therefore, the identification and characterization of critical genes responsible for tumor progression and metastasis is the focus of numerous investigations all over the world (Siegel, Giaquinto & Jemal, 2024; Suhail et al., 2019). Multiple strategies have been used to improve the cancer patient’s life, including surgery, radiation or chemical therapy and so on. CRISPR-based strategies to edit somatic mutations and chimeric antigen receptor T cells (CAR-T) engineered to target tumor antigens like CD19. Other approaches involve RNA therapeutics (miRNA, siRNA) and oncolytic viruses modified to selectively replicate in tumors. Therefore, it is important to further investigated the molecular mechanism of cancer metastasis (Sonkin, Thomas & Teicher, 2024).

Metastasis associated proteins (MTAs) are a group of transcriptional co-regulators, of which, metastasis-associated antigen 1 (MTA1) was originally cloned from mouse metastatic tumor tissues by Toh, Pencil & Nicolson (1994) on chromosomes 14q32, while MTA2 gene was identified at chromosome 11q12-13.1 using fluorescence in situ hybridization (Futamura et al., 1999) and MTA3 was location at chromosome 2p21 (Fujita et al., 2003). MTA1, MTA2 and MTA3 form distinct biochemically protein complexes and function as integral subunit of the nucleosome remodeling and deacetylation (NuRD) complex (Xue et al., 1998; Yao & Yang, 2003). Their roles in chromatin assembly, transcription, genomic stability and the pathologic states are distinguishing (Smits et al., 2013; Zhang et al., 1999). Although both MTA1 and MTA2 are among the most upregulated in human cancers (Kumar & Wang, 2016), their functions do not always overlap. MTA3, which firstly discovered in breast cancer is considered to inhibit invasive growth pathway and the expression of which is dependent on estrogen action (Kumar, Wang & Bagheri-Yarmand, 2003). Thus, it is crucial to discuss the expression and function of MTA proteins respectively. As the diverse functions of MTA1 and its role in various cancers have been discussed continuously (Malisetty et al., 2017; Toh & Nicolson, 2014). In this review, we examine the multifaceted biological roles of MTA2 and summarize the mechanism of the properties in various biological and pathological pathways which establishes transcriptional modulation of a number of target genes.

MTA2 is a 668-amino-acid protein and identified as a protein highly related (65% identical) and shorter than MTA1 (Zhang et al., 1998). The human MTA2 gene has 20 exons and seven transcripts, and the MTA2 protein contains four distinct domains, including the BAH, EML2, SANT and an atypical zinc-finger domain (UniProt, 2014) (Fig. 1). These structural domains provide internal clues about the potential MTA2 functions. Using Gel-mobility-shift and mutagenesis studies, Xia & Zhang (2001) revealed that transcription factor Sp1 and ETS elements play important roles in regulating mouse Mta2 transcription. Interestingly, the target genes controlled coordinately by Sp1 and ETS family transcription factors are all up regulated in tumors, and the proteins encoded by these genes are in some form linked to cancer metastasis (Xia & Zhang, 2001). The Sp1/ETS-MTA2 axis may provide a novel therapy target for oncology treatment. Except being in the presence of NuRD complex, MTA2’s suspected role was also involved in a HDAC1-containing complex, which also known as PID, with MTA2 and HDAC2 in mediation the deacetylation of p53 (Luo et al., 2000). Using size exclusion chromatography and negative stain electron microscopy, Brasen et al. (2017) purified and structural characterized the human full-length MTA2-RBBP7 complex. In the absence of HDAC1, the MTA2-RBBP7 complex appears capable of hinge like motion around its center. The expression and fundamental function of MTA2 proteins in cell program process.

Figure 1 Schematic structure of the MTA2 protein.

MTA2 a 668-amino-acid protein which contains a highly conserved one each of BAH domain, ELM domain, SANT domain and GATA-like zinc finger (ZnF) domain.

Over the past several decades, extensive research has established that the MTA2 protein plays key roles in the the regulation of gene expression, maintenance of genome stability (Errico, Aze & Costanzo, 2014), DNA replication (Christov et al., 2018), and embryonic development (Laugesen & Helin, 2014; You et al., 2021). A schematic overview of these functions is presented in Fig. 2.

Figure 2 The main functions of MTA2 in development.

MTA2 is involved in diverse physiological processes such as DNA replication, damage repair, apoptosis, autophagy, cell cycle, T lymphocytes proliferation, B cell development, replication fork integrity, nerve system and preimplantation development. Created with BioRender.com.

In the embryo of Xenopus laevis, MTA2, as a core component of the NuRD complex, compensates for the functional requirement of Y RNAs for DNA replication during early embryogenesis (Christov et al., 2018). MTA2 is essential for the recruitment of the replisome-associated protein Tipin to chromatin, and its depletion leads to a significant reduction in the binding of polymerase α (Pol α) to DNA. MTA2, together with Tipin, contributes to the preservation of replication fork integrity and efficiency, which appears to depend on the role of the MTA2/NuRD complex in maintaining heterochromatin structure (Errico, Aze & Costanzo, 2014). In chromatin regions containing DNA double-strand breaks (DSB), knockdown of MTA2 and CHD4, another core subunit of NuRD complex, lead to accumulation of spontaneous DNA damage and increase ionizing radiation (IR) sensitivity by laser micro irradiation (Smeenk et al., 2010). The BMRF1 protein of Epstein-Barr virus localize with NuRD components at viral replication compartments using the motif in the BMRF1 transcriptional activation sequence. BMRF1 acts at the same step in the DNA damage countermeasure as MTA2/NuRD complex, suggesting that it interferes with NuRD function in response to double-stranded DNA breaks (Salamun et al., 2019). Stanniocalcin 2 (Stc2) is a newly identified target gene of the aryl hydrocarbon receptor (AhR), whose expression is responsive to the endogenous AhR agonist cinnabarinic acid, transcription factor AhR could interact with MTA2 in cinnabarinic acid-dependent manner and lead to MTA2 recruitment to the Stc2 promoter, which results in cryoprotection about liver cells exposed to chemical insults (Joshi, Hossain & Elferink, 2017). MTA2 is identified as a chromatin binding protein to recruit the methyltransferase EZH2 to silence targeted genes including tuberous sclerosis 2 (TSC2), which in turn modulates subsequent MTOR pathway, leads to the inhibition of autophagy (Wei et al., 2015). In human primary keratinocytes, a functional association between the LINC00941 and the NuRD complex has been observed, which contributes to the repression of *EGR3*, a key transcription factor involved in keratinocyte differentiation. This regulatory mechanism prevents premature differentiation and maintains cellular homeostasis (Morgenstern et al., 2024).

Rationale

MTA2, a key transcriptional regulator, plays pivotal roles in normal development and cancer, yet its physiological functions and pathogenic mechanisms remain incompletely understood. Existing research is fragmented, with conflicting findings on its proliferative effects in solid tumors, hindering translational progress. This review is critical as it synthesizes scattered evidence, bridging MTA2’s roles in normal processes (embryonic development, immune cell differentiation) and cancer biology. It clarifies its oncogenic mechanisms, including aberrant amplification, associations with metastasis, and regulation by non-coding RNAs/transcription factors, resolving inconsistencies. Highlighting MTA2’s correlation with poor prognosis and its potential as a therapeutic target addresses a gap in precision medicine. By identifying unresolved questions, such as context-dependent immune roles and clinical translation barriers, it guides future research. This integration of basic and clinical insights fills a vital niche, advancing understanding of MTA2’s multifaceted roles to inform novel diagnostics and therapies.

Survey Methodology

We searched PubMed, the Web of Science, Embase, and Cochrane. These platforms provide extensive access to peer-reviewed articles, ensuring coverage of a wide range of studies across disciplines. Search Terms: “MTA2”, “caner”, “metastasis” and “normal development” are searched in combination with the subject title or its free term respectively. Inclusion criteria: Articles were selected based on the following: Studies detailing the mechanisms of MTA2 in cancer, including Peer-reviewed journal articles and reviews. Exclusion criteria: Studies lacking robust methodologies or clear conclusions. Screening process: Titles and abstracts were reviewed to ensure relevance, followed by a detailed examination of the full text.

The function of MTA2 in immune cells

Mta2 null mice exhibits partial embryonic lethality, while the surviving mice develop lupus-like autoimmune symptoms. Mta2 null T lymphocytes show hyperproliferation upon stimulation, which correlates with hyper-induction of interleukin (IL)-2, IL-4 and interferon (IFN)-γ (Lu et al., 2008). During mice preimplantation development, Mta2 is the only zygotically expressed Mta gene prior to the blastocyst stage. Knockdown of Mta2 leads to biallelic H19 expression and loss of DNA methylation on the imprinting control region in blastocysts (Ma et al., 2010). During T helper cell differentiation, MTA2 acts as a critical player in T cell mediated immunity. Hwang et al. (2010) using affinity purification and mass spectrometry verified that transcription factor GATA3 interacts with MTA2 binding to several regulatory regions in the Th2 cytokine locus and the ifng promoter. In mature T cells, by repressing NK cell-associated transcription, transcription factor BCL11B and the NuRD complex bind to each other to maintain T-cell fate directly (Liao et al., 2023).

In B cell development including in pro-B, pre-B, immature B, marginal zone B cells and abnormal germinal center B cell differentiation during immune responses, MTA2 deficiency in mice leads to increased H3K27 acetylation at both Igll1 and VpreB1 promoters (Lu et al., 2019). MTA2/NuRD complex interacts with AIOLOS/IKAROS in pre-B cells and cooperates with OCA-B in the pre-B cells to immature B cells transition.

In erythroid cells, transcription factor GATA1 could recruit the repressive MeCP1 complex (including MTA2) and the chromatin remodeling ACF/WCRF complex to act as an activator or repressor of different target genes respectively (Rodriguez et al., 2005).

The function of MTA2 in nerve system

Chromodomain helicase DNA binding protein 4 (Chd4), the core catalytic subunit of NuRD complex, along with Mta2, is recruited to genes which are either positively and negatively regulated by Egr2 during peripheral nerve myelination (Hung, Kohnken & Svaren, 2012). The transcription factor A+U-rich element binding factor 1 (AUF1) plays a role in integrating the genetic and epigenetic signals through recruiting HDAC1 and MTA2 to AT-rich DNA elements in developing cortical neurons (Lee et al., 2008).

The expression and clinical significance of MTA2 in cancer

Metastasis is widely recognized as the leading cause of cancer-related mortality and remains the ultimate challenge in oncology (Steeg, 2016). The molecular mechanisms of metastatic tumor cells to infiltrate the surrounding tissue are different, an accurate description and characterization about the movement of tumor cells from primary site to progressively distant organs and colonization that underlie this multistep process is the critical point (Fidler & Kripke, 2015; Nieto, 2013). Among the complicated genes that underlie this multistep process, the metastasis-associated antigen has been shown to work as significant factors to play distinct roles in this dynamic event (Covington & Fuqua, 2014; Malisetty et al., 2017). In particular, MTA1 and MTA2 are frequently upregulated genes in various human cancers and act as pivotal drivers of metastasis. Their functional roles, however, appear to be cancer-type-specific, operating through distinct molecular mechanisms (Kumar & Wang, 2016). Based on the overview of both historic and recent experimental evidence, we dissect the critical roles of MTA2 in tumor progress, the relative schematic diagram is shown in Fig. 3.

Figure 3 MTA2 is involved in different cancer types.

Ambiguous functions of MTA2 have been described in breast cancer, gastric cancer, pancreatic cancer, hepatocellular carcinoma, colorectal cancer, glioma, renal cell carcinoma, cervical cancer, ovarian cancer, nasopharyngeal carcinoma, oral cancer, esophageal carcinoma, non-small cell lung cancer, thyroid cancer and thymomas. Created with BioRender.com.

Numerous studies on MTA2 function suggest its involvement in cancer progression is likely mediated through its frequently gene amplification and its interactions with a variety of chromatin remodeling factors (Lai & Wade, 2011). MTA2 acts as a central hub with large number of regulatory genes that are involved in different signaling pathways and may work as a molecular marker in various solid tumors (Fu et al., 2011; Kumar, Wang & Bagheri-Yarmand, 2003). The key roles of MTA2 in different cancer types are summarized in Table 1.

Table 1 The primary features and molecular mechanism of MTA2 in different cancer types.

Cancer type	Cell line	MTA2 effect	Clinical
application	Molecular targets	Pathway	Interaction protein	Upstream regulation	Reference	
Breast cancer	Mouse 4T1	Migration and invasion	Lung metastasis in mice	E-cadherin		TWIST/NuRD		Yang et al. (2004)	
Breast cancer	MCF-7			pS2
c-myc			OHT-ER	Liu et al. (2024)	
ERa-negative breast
cancer	MDA-MB-231	Migrate and survival	Metastasis
prognostic for early recurrence	Rho pathway
such as RhoA-C, focal adhesion kinase, and Rho Kinase (ROCK)		NuRD complex		Covington et al. (2013)	
ERa-positive
breast cancer	MCF-7,
T47D, MDA-MB-361, BT474	Anchoragein
dependent growth	Metastasis,
predictive biomarker,
a therapeutic target of ERα	The deacetylation of ERα protein,
Repressor of
ERα activity		NuRD complex		Cui et al. (2006)	
ER positive breast cancer	MCF-7	Epithelial-mesenchymal transition, metastasis		Repression
E-Cadherin		AIB1		Vareslija et al. (2021)	
Triple-negative breast cancer	MDA-MB-231, 4T1 cells	Angiogenesis		SerRS	VEGFA	NuRD complex	3-(4-methoxyphenyl) quinolin-4(1H)-one (MEQ)	Zhang et al. (2020)	
Gastric cancer	AGP01 cells		Advanced GC stages (tumor invasion, lymph nodes metastasis, Distant
metastasis)				MYC	Lopes et al. (2019)	
Gastric cancer			Tumor invasion, T staging				Sp1	Zhou et al. (2012)	
Gastric cancer	SGC-7901 and AGS	Invasion and metastasis,
xenograft growth					Sp1	Zhou et al. (2013)	
Gastric cancer	BGC-823 and MKN28	Colony formation,
Tumor growth		IL-11				Putoczki et al. (2013)	
Gastric cancer	MKN-45, SGC-7901,
MGC-803	Epithelial–mesenchymal transition, migration and metastasis			PI3K/
Akt		miR-1236-3p	An et al. (2018)	
Gastric cancer	SGC-7901	Cell proliferation, migration and invasion	Tumor invasion,
node and metastasis grade, tumor embolus
formation,	KAI-1, E-cadherin			Long non-coding RNA SNHG5	Zhao et al. (2016)	
Pancreatic cancer	MIA Paca-2 and
PANC-1	Cell proliferation, migration and invasion	Advanced stage, poorer
prognosis	PTEN	PI3K/
Akt	NuRD complex	Snail	Nishioka et al. (2010)	
Pancreatic cancer			Poorer tumor differentiation, TNM stage, lymph node metastasis					Chen et al. (2013)	
Pancreatic cancer	PANC-1 and BxPC-3	Invasion and proliferation	Shorter
overall survival time	E-cadherin		HDAC1/NuRD complex	HIF-1α	Zhu et al. (2018)	
Pancreatic cancer	Colo357 and Panc-1	Invasion			IRAK-1/NF-κB signaling		miR-146a	Li et al. (2010a) and Li et al. (2010b)	
Pancreatic cancer	PANC-1, BxPC-3 and SW1990	Cell proliferation and invasion	Shorter survival time	HIF-1α			lncRNA-MTA2TR,
ATF3	Zeng et al. (2019)	
Hepatocellular carcinoma			Tumor size and differentiation,
a predictor of aggressive phenotypes					Lee et al. (2009)	
Hepatocellular carcinoma	SK-Hep-1 and Huh-7	Migration and invasion,
no
significant effect on HCC cell growth	Tumor grade, overall survival of HCC
patients	Matrix
metalloproteinase 2 (MMP2)	p38MAPK/MMP2			Hsu et al. (2019)	
Hepatocellular carcinoma	HepG2 cells	The proliferation and growth	Advanced pathological stages,	FRMD6	Hippo signaling pathway			Guan et al. (2019)	
Colorectal cancer	HCT116
cells	Cells growth					Histone acetyltransferase p300	Zhou et al. (2014)	
Colorectal cancer	SW480 and SW620 cells	Migration					β-defensin-3	Uraki et al. (2014)	
Glioma	GBM8401 and Hs683	Growth, cell migration and
invasion	Tumor grade					Cheng et al. (2014)	
Glioma	U87 and U373	Proliferation and invasion					miR-548b	Pan et al. (2016)	
Renal
cell carcinoma	786-O, Caki-1, and ACHN	Migration, invasion, and in vivo metastasis
did not affect the proliferation	Tumor grade	Inhibition of miR-133b, thus activate
matrix metalloproteinase-9				Chen et al. (2019)	
Oesophageal squamous cell carcinoma	KYSE-30
KYSE-510	Proliferation,
migration and invasion, EMT	Increased expression in the ESCC samples	E-cadherin			Small nucleolar RNA host gene 5 (SNHG5)	Wei et al. (2020)	
Oesophageal squamous cell carcinoma	KYSE30 and KYSE510	ESCC growth,
metastasis, and epithelial-mesenchymal transition (EMT)	The malignant characteristics and poor prognosis	Positively regulates the expression
of eukaryotic initiation factor 4E (EIF4E) and Twist, inhibition of E-cadherin		EIF4E, Twist		Dai et al. (2020)	
Cervical cancer			Federation of gynecology and obstetrics (FIGO) stage and lymph node metastasis, poor
overall survival time					Xiao et al. (2016)	
Cervical cancer	HeLa,
SiHela and C33A	Cell migration and invasion, lung
metastasis	Advanced
Tumor Grade and Poor Survival	Transcriptional Suppression of miR-7	SP1/Kallikrein-10 (KLK10) axis		(transcription factor specificity
protein 1) Sp1	Lin et al. (2020)	
Cervical cancer	HeLa, SiHa, and C33A	Migration, invasion, lung metastasis, but no effect on proliferation	Poor survival	AP1
transcriptional activity and MMP12	ASK1/MEK3/p38			Lin et al. (2021)	
Ovarian epithelial cance			Clinical stage, histopathological grade and lymph node metastasis					Ji et al. (2006)	
Oral cancer	OECM1, HSC3, and SAS
cells	Cell migration and
invasion, but not significantly affect cell proliferation	Tumor grade,
the overall survival rate of
patients with grade III tumor	p-cofilin and LC3-II expression				Tseng et al. (2019)	
Oral squamous cell carcinoma	Cal27, HN4	The migration,
invasion and EMT	T category, N category,
TNM stage, and histological grade, survival probability,				HOX antisense intergenic RNA (HOTAIR)/miR-326 axis	Tao et al. (2020)	
Non-small cell lung cancer			Advanced TNM stages, tumor size, lymph node metastasis					Liu et al. (2012)	
Lung cancer	A549	Inhibited primary lung cancer growth but later favored metastasis,
epithelial mesenchymal transition, and lung tumor metastasis		Down regulation of CHD1, KRT18; up regulation of FN1, VIM	Activation of NF-κB signaling	NuRD	IKK2	El-Nikhely et al. (2020)	
Non-small cell lung cancer			Positive correlation with clinical stage and lymph node metastasis, negative correlation with differentiation degree of NSCLC					Wang et al. (2010)	
Non-small-cell lung cancer	A549 and SPC-A-1 cells	Invasion	Lymphnode metastasis				d-Catenin/Kaiso	Dai et al. (2011)	
Thymomas			Histological type and Masaoka stage					Wang et al. (2012)	
Nasopharyngeal carcinoma	CNE2 and
C666-1 cells	Cell proliferation,
migration, and invasion					miR-148b	Wu et al. (2017)	
Nasopharyngeal carcinoma	CNE1,
CNE2, and HNE1	Proliferation and invasion	Clinical stage and lymph node metastasis	Upregulated the expression of matrix metalloproteinase 7 and cyclin D1	Akt activity			Wu et al. (2016)	
Papillary thyroid cancer	TPC-1 and SW579	Cell proliferation,
and invasion					circ-NCOR2/ miR-615a-5p	Luan et al. (2020)	

Breast cancer

In breast cancer, the function roles and regulatory mechanisms of MTA2 have been extensively investigated. In breast tumor cell line MCF-7, components of NuRD complex, including MTA1/2, are recruited by a 4-hydroxytamoxifen (OHT)-bound estrogen receptor (ER) to the target gene promoters such as pS2 and c-myc (Liu & Bagchi, 2004). Using purification and analysis of the TWIST protein complex by mass spectrometry, TWIST could interact with several components of the Mi2/NuRD complex such as MTA2, RbAp46, Mi2, HDAC2 and recruit them to the E-cadherin promoter for transcriptional repression. Given that TWIST functions as a master regulator of EMT and breast cancer metastasis (Yang et al., 2004), the TWIST/Mi2/NuRD complex has been shown to be essential for breast cancer cell migration and invasion in vitro, as well as for lung metastasis in mice (Fu et al., 2011). Si et al. (2015) provided more evidence to show the difference and relationship between MTA1 and MTA2 in breast cancer cells. We identified GATA3, which is the most highly expressed transcription factor in the luminal epithelial cell population, could form a G9A/NuRD (MTA3) complex to target a cohort of genes including ZEB2. While through the recruitment of G9A/NuRD (MTA1) complex, ZEB2 could in turn, repress the expression of G9A and MTA3. Although the existence of ZEB2/G9A/NuRD (MTA2) complex in vivo was confirmed, MTA2/NuRD complex was not involved in the molecular basis for the opposing action of MTA3 and MTA1 in breast cancer progression (Si et al., 2015).

Breast cancer is a highly heterogeneous malignancy that can be classified into different subtypes based on histological and molecular characteristics. These subtypes include luminal A, luminal B cancers, HER2 positive (HER2+), basal-like and triple-negative breast cancers (Eroles et al., 2012). The role of MTA2 and its potential underlying mechanisms have been investigated in carious contexts. In ERα positive breast cancer, MTA2 has been shown to promote anchorage-independent growth and serve as a potential predictive biomarker. This effect is mediated through the binding of MTA2 to ERα, which subsequently suppresses the transcriptional activity of ERα (Cui et al., 2006). Through interaction with the coactivator AIB1, MTA2 could form a repressive complex, inhibiting CDH1 (encoding E-cadherin) to promote EMT and associate with a pro-metastatic phenotype in ER positive breast cancer metastasis (Vareslija et al., 2021). In ERα-negative breast patients, MTA2 expression is associated with poor prognostic markers and increased risk of early recurrence in retrospective analyses through activation of the Rho signaling pathway through activation of the Rho signaling pathway (Covington et al., 2013). In luminal B breast cancer cells ZR-75-30, the overexpression of MTA1 could down regulate the intrinsic inhibitor of the neutrophil elastase (NE), elafin, to promote the degradation of MTA2, therefore inhibiting the metastasis of ZR-75-30 cells in vitro. They declared the opposite role of MTA1 and MTA2 in the metastasis of ZR-75-30 cells in vitro (Zhang et al., 2019). In triple-negative breast cancer, a cell-based seryl tRNA synthetase (SerRS) promoter driven dual luciferase reporter system is used to screen and verify that 3-(4-methoxyphenyl) quinolin-4(1H)-one (MEQ), is an isoflavone derivative to increase SerRS expression and a potent transcriptional repressor of VEGFA. MEQ regulated SerRS transcription by interacting with MTA2 to suppress the angiogenesis in TNBC allografts and xenografts in mice (Zhang et al., 2020). As breast cancer is a highly heterogeneous cancer, in the future, it is better for the researchers to understand the role of MTA2 and its related mechanisms in the progression of different breast cancer type, the relative schematic diagram is shown in Fig. 4.

Figure 4 Working model of MTA2 in breast cancer.

The function and potential targets of MTA2 in different breast cancer subtypes such as ERa-positive, ERa-negative, Luminal B, and triple-negative breast cancer are shown. Created with BioRender.com.

Gastric cancer

Gastric cancer ranks as the fifth most commonly diagnosed malignancy worldwide (Bray et al., 2018). Numerous studies have demonstrated that the expression of MTA2 significantly associated with key clinicopathological features of gastric cancer, including tumor invasion, lymph nodes metastasis and tumor node metastasis (TNM) staging (Zhou et al., 2013). Between early-stage gastric cancer tissues of M0 and M1 patients, the mRNA and protein levels of MTA2 are significantly different (Lopes et al., 2019). Transcription factor specificity protein 1 (Sp1) is found to be overexpressed and had positive correlation with MTA2 (Zhou et al., 2012). Furthermore, silencing MTA2 has been shown to markedly inhibit gastric cancer cell invasion both in vitro and in vivo. Mechanistically, Sp1 can bind to the MTA2 promoter region and enhance its transcriptional activity, potentially through upregulating the expression of CD24 and MYLK (Zhou et al., 2013).

Helicobacter pylori (H. pylori) is the main environmental factor and the main causes of gastric cancer (Yang et al., 2020). CircRNAs are universal endogenous noncoding RNAs with highly conserved and stable covalently closed cyclic structure and can act as miRNA sponges to inhibit the activity of miRNAs (Jeck & Sharpless, 2014). In AGS cells, using RNA-seq analysis, circMAN1A2 is proved as one of the upregulated circRNAs after infected with Hp26695. They further revealed that H. pylori could induce circMAN1A2 expression to promote the carcinogenesis of gastric cancer by sponging miR-1236-3p to regulate MTA2 expression in vitro and in vivo (Guo et al., 2022). circMTA2 is another circular RNA which interacted with ubiquitin carboxyl-terminal hydrolase L3 (UCHL3) to restrain MTA2 ubiquitination, and thereby facilitating tumor progression though stabilize MTA2 protein expression (Xie et al., 2024).

In gastric cancer cell lines BGC-823 and MKN28 with MTA2 overexpression, MTA2 has been shown to promote colony formation and tumor growth through the regulation of IL11. However, the mechanism by which the HDAC inhibitor SAHA suppresses IL11 expression remains unclear (Putoczki et al., 2013; Zhou et al., 2015). MicroRNAs are a class of small, evolutionarily conserved non-coding RNAs, approximately 19–25 nucleotides in length, that function as key regulators in cancer development. They can act either as oncogenes or tumor suppressors depending on their target genes and cellular context (Ohtsuka et al., 2015). miR-1236-3p has been identified as a tumor suppressor in gastric cancer by directly targeting MTA2, thereby inhibiting its expression and suppressing the activation of the PI3K/Akt signaling pathway (An et al., 2018). Long non-coding RNAs are a class of RNA transcripts which are longer than 200 nucleotides and involved in gastric cancer development (Ghafouri-Fard & Taheri, 2020; Okugawa et al., 2014), among which, the long non-coding RNA SNHG5 prevents the translocation of MTA2 from the cytoplasm into the nucleus, and thereby interfering with the formation of the NuRD complex to suppresses gastric cancer progression (Zhao et al., 2016), the relative schematic diagram is shown in Fig. 5A.

Figure 5 Working model of MTA2 in gastric cancer and pancreatic cancer.

(A) In gastric cancer, MTA2 is positive regulated by SP1 and negative regulated by IncRNA SNHG5 or miR-1236-3p, through targeting CD24, MYLK, IL-11, MTA2 promotes invasion, mobility, proliferation and function as a biomarker for poor prognosis. (B) In pancreatic cancer, MTA2 is positive regulated by HIF-1α, IncRNA MTA2TR and negative regulated by miR-146a, through targeting E-cadherin and PTEN, MTA2 is involved metastasis, proliferation and could work as a biomarker for poor prognosis and therapeutic target.

Pancreatic cancer

Pancreatic cancer (PC) is recognized as a highly lethal malignant, characterized by aggressive biological behavior in recent decades (Hessmann et al., 2017). Despite significant efforts to develop novel therapeutic strategies, the clinical outcomes remain poor, with 5-year survival rates of less than 7% (Pavlidis & Pavlidis, 2018; Waddell et al., 2015). In pancreatic ductal adenocarcinoma, elevated expression of metastasis-associated protein 2 (MTA2) has been observed in tumor tissues. This overexpression is associated with shorter overall survival and has been identified as an independent prognostic factor. Mechanistically, MTA2 promotes PDAC cell proliferation and invasion in vitro, as well as tumor growth in vivo, through its repressive binding to the promoter region of phosphatase and tensin homolog (PTEN) (Si et al., 2019). Snail family transcriptional repressor 1 (Snail), which is a master regulator of epithelial-mesenchymal transition (EMT) and metastasis (Chen et al., 2014; Nishioka et al., 2010), could recruit MTA2 and HDAC1 to suppress PTEN expression and thus activate the PI3K/Akt pathway. These data indicated that in PDAC cancer, MTA2 might work as a subunit of NuRD complex in promotion of PDAC progress. Consistent with the above conclusion, in the study from Chen et al. (2013) they also found the mRNA and protein expression levels of MTA2 are both significantly upregulated in PDAC lesion. The higher MTA2 expression is correlated with poorer tumor differentiation, TNM stage, lymph node metastasis and is considered as a prognostic marker for PDAC. In PDAC cancer cells, MTA2 is transcriptionally upregulated by HIF-1α through an hypoxia response element (HRE) of the MTA2 promoter in response to hypoxia, reciprocally, MTA2 could deacetylate HIF-1α and enhance its stability through interacting with HDAC1 to promote the progression and metastasis of pancreatic carcinoma (Zhu et al., 2018).

Emerging evidence underscores the pivotal roles of miRNAs and lncRNAs in the regulation of PDAC metastasis (Nicoloso et al., 2009). Notably, miR-146a has been found to be downregulated in PDAC cell lines. Treatment of PDAC cells with B-DIM and G2535 has been shown to upregulate expression of miR-146a expression, leading to the suppression of key downstream targets including EGFR, MTA-2, IRAK-1, and NF-κB, thereby inhibiting cell invasion (Li et al., 2010a). Furthermore, re-expression of miR-146a in PC cells results in the attenuation of IRAK1/NF-κB signaling and reduced MTA2 expression (Li et al., 2010b). These findings collectively highlight the tumor-suppressive function of miR-146a in PDAC progression and suggest its potential as a therapeutic target.

lncRNA-MTA2TR (MTA2 transcriptional regulator RNA, AF083120.1) is overexpressed in PC patient tissues and transcriptionally upregulates MTA2 by recruiting activating transcription factor 3 (ATF3) to the MTA2 promoter region. Under hypoxic conditions, MTA2TR is transcriptionally regulated by HIF-1α. A positive feedback loop involving MTA2TR, MTA2 and HIF-1α may play a critical role in regulating of tumorigenesis (Zeng et al., 2019). A schematic representation of MTA2 in PDAC is shown in Fig. 5B.

Hepatocellular carcinoma

In hepatocellular carcinoma, the MTA2 expression level is strongly increased depending on the tumor size and differentiation, and might be a predictor of aggressive phenotypes (Lee et al., 2009), however, the molecular mechanism of MTA2 expression in HCC need further investigation. MTA2 silencing drastically reduces migration and invasion capability through inhibits matrix metalloproteinase 2 (MMP2) and decreases the phosphorylation of the p38MAPK protein (Hsu et al., 2019). As Guan et al. (2019) reported, using ChIP-seq analysis, they identified MTA2 represses a cohort of transcriptional targets including FRMD6, which is a key upstream component of Hippo signaling pathway. Via repressing FRMD6, MTA2 could promote HCC progression through Hippo pathway. Protein tyrosine kinase 7 (PTK7) acted as a downstream factor for MTA2 expression recombinant matrix metalloproteinase 7 (MMP7) reversed the PTK7 knockdown-induced suppression of migration and invasion in HCC cells, which indicate MTA2-FAK-MMP7 axis might be a diagnostic value for HCC patients (Hu et al., 2024). In CD133+ HCC cells, MTA2 could interact with HDAC2/CHD4, forms by a part of the NuRD complex and transcriptionally inhibits BDH1, a major rate-limiting enzyme in the metabolic process of ketone bodies, controls the transformation between acetoacetic acid (AcAc) by R-loops, leading to the accumulation of βHB, the increase in H3K9bhb, and a waterfall effect on HCC formation and progression (Zhang et al., 2021).

Colorectal cancer

In colorectal cancer cells, histone acetyltransferase p300 could bind to MTA2 and acetylate MTA2 at K152, while MTA2 acetylation mutation could inhibit the colorectal cancer cells growth and Rat1 fibroblasts invasion capacity. However, the expression level of MTA2 in colorectal cancer tissues and cell lines need further investigation (Zhou et al., 2014). Human β-defensins (hBDs) have chemotactic activity for memory T cells and immature dendritic cells with certain roles in cancer (Sorensen et al., 2005). In colon cancer cells, hBDs could reduce the expression of MTA2 and inhibit the migration ability in a paracrine fashion (Uraki et al., 2014).

Glioma

In glioma tumor tissues, the expression of MTA2 was significantly correlated with tumor grade, while knockdown of MTA2 could significantly inhibit glioma cells growth and invasion in vitro and in vivo (Cheng et al., 2014). MTA2 was considered as a direct target of miR-548b, through repression of MTA2, miR-548b exerts its tumor suppression function in glioma (Pan et al., 2016).

Renal cell carcinoma

During the progression of renal cell carcinoma (RCC), MTA2 expression is markedly upregulated in RCC tissues and correlates with higher tumor grade. Knockdown of MTA2 has been shown to decrease both the activity and protein levels of matrix metalloproteinase-9 (MMP-9), leading to the suppression of RCC cell migration, invasion, and in vivo metastasis. One potential molecular mechanism by which MTA2 contributes to RCC metastasis involves the regulation of miR-133b expression, which directly targets MMP-9. Notably, this regulatory pathway does not appear to influence cancer cell proliferation (Chen et al., 2019).

Cervical cancer

In cervical cancer, elevated expression of MTA2 also indicates poor prognosis of cervical cancer patients (Xiao et al., 2016). Lin et al. (2020) revealed the potential mechanism as that knockdown of MTA2 could elevate the expression of miR-7 and then by direct inhibition of transcription factor specificity protein 1 (Sp1) expression. The enhanced KLK10 expression works as a downstream target gene. Through negatively correlated with Kallikrein-10 (KLK10) expression in vitro and in vivo, they provide a potential therapeutic target in cervical cancer (Lin et al., 2020). As reported by Lin et al. (2021), MTA2 is highly expressed in cervical cancer cells, through regulate matrix metalloproteinase 12 (MMP12) expression, MTA2 is involved in the lung metastasis of cervical cancer.

Ovarian cancer

In ovarian epithelial cancer, MTA2 expression levels in malignant ovarian tissues are significantly elevated compared to those in normal epithelial tissues, and this upregulation is associated with advanced clinical stage, histopathological grade and lymph node metastasis (Ji et al., 2006). Functional analyses of recombinant MTA1 and related MTA2 proteins suggest that the MTA1 protein possesses histone deacetylase activity, highlighting the potential role of MTA family proteins in promoting cancer cell invasion and proliferation (Nicolson et al., 2003).

Nasopharyngeal carcinoma

In nasopharyngeal carcinoma (NPC), MTA2 is identified as a direct target of miR-148b, the tumor suppressive effects of miR-148b is partly through suppression of MTA2 (Wu et al., 2017). However, although MTA2 is associated with malignant behaviors of several types of tumor cells including NPC cell lines, whether MTA2 is indeed involved in the progression of NPC still need further discussion (Wu et al., 2016).

Oral cancer

In human oral cancer, the expression of MTA2 is significantly upregulated in oral cancer tissues and cell lines compared to non-tumor oral tissues and cell lines. Mechanistically, knockdown of MTA2 has been shown to inhibit cell migration and invasion, potentially through the modulation of cytoskeletal dynamics and autophagy, as evidenced by increased expression of phosphorylated cofilin (p-cofilin) and microtubule-associated protein 1 light chain 3-II (LC3-II) (Tseng et al., 2019). In oral squamous cell carcinoma (OSCC), HOX antisense intergenic RNA (HOTAIR) is markedly overexpressed in both tumor tissues and cell lines. HOTAIR functions as a competitive endogenous RNA (ceRNA), effectively sponging miR-326 and thereby relieving the post-transcriptional suppression of MTA2. Elevated MTA2 expression is significantly correlated with advanced clinicopathological features and poor prognosis for patients with OSCC (Tao et al., 2020).

Esophageal carcinoma

In esophageal squamous cell carcinoma (ESCC), MTA2 has been identified as a binding partner of small nucleolar RNA host gene 5 (SNHG5) through RNA pull down assays. Overexpression of SNHG5 results in the downregulation of MTA2 at the transcriptional level and promotes its ubiquitin-mediated proteasomal degradation. Furthermore, the mRNA expression of MTA2 is significantly elevated in ESCC specimens, the negative correlation between SNHG5 and MTA2 might provide a new potential therapeutic strategy for ESCC (Wei et al., 2020). Similarly, Moreover, MTA2 is frequently overexpressed in ESCC tissue and is closely associated with aggressive tumor phenotypes and poor prognosis. Mechanistically, MTA2 promotes tumor growth, metastasis, and epithelial-mesenchymal transition (EMT) via a novel positive feedback loop involving eukaryotic translation initiation factor 4E (EIF4E) and Twist (Dai et al., 2020).

Non-small cell lung cancer

In non-small cell lung cancer (NSCLC), MTA2 is predominantly localized in both nucleus and cytoplasm in cancer cells. Notably, a higher Ki-67 proliferation index has been significantly associated with nuclear MTA2 positive tumors. However, there is no significant difference in cytoplasmic MTA2 status with Ki-67 proliferation index (Liu et al., 2012). MTA2 has been identified as a non NF-κB target gene regulated by IKK2, MTA2 negatively regulates NF-κB signaling to reduce lung tumor growth and inflammation. MTA2 exerts opposite functions as it initially inhibited primary lung cancer growth but later favored metastasis by sustained inflammatory response at the later phases (El-Nikhely et al., 2020). In NSCLC, the expression of MTA2 has negative correlation with differentiation degree, but positive correlation with clinical stage and lymph node metastasis (Wang et al., 2010). Furthermore, upregulation of δ-Catenin, an adherens junction-associated protein, may regulate MTA2 through Kaiso—a transcription factor—in a DNA methylation-dependent manner, leading to adverse clinical outcomes in non-small cell lung cancer (NSCLC) (Dai et al., 2011).

Thyroid cancer

In papillary thyroid cancer (PTC), circular RNA circ-NCOR2 has been found to be upregulated both PTC tissues and cell lines. It promotes PTC cell progression by enhancing MTA2 expression through sponging miR-615-5p (Luan et al., 2020).

Thymomas

In thymomas, nuclear MTA2 is detected in 70.8% of the thymomas, there are good consistencies and correlations between cytoplasmic p120 catenin, cytoplasmic Kaiso and nuclear MTA2 expression. These three protein correlated directly with histological type and Masaoka stage of thymomas, and might be used as potential biomarkers to predict the biological behavior (Wang et al., 2012). However, the mechanism underlying the relationship between MTA2 and Kaiso remains unknown.

Discussion

Many studies used TCGA data to explore MTA2 genes in cancers, however, TCGA exhibits inherent technical and biological biases. Technically, batch effects from multi-institutional protocols, normalization artifacts due to unaddressed tumor heterogeneity and arbitrary filtering of low-expression genes risk omitting critical tumor-related signals. Biologically, single-timepoint sampling fails to capture tumor evolution dynamics, while anatomic selection bias overrepresents easily accessible tumors (Liu, Guo & Wang, 2024). Bulk RNA-seq further conflates tumor-intrinsic signals with stromal/immune microenvironment noise, and demographic underrepresentation limits biomarker generalizability (Liu et al., 2025). These compounded biases necessitate multimodal validation, and diverse cohort designs to refine translational relevance.

Considering various studies demonstrating the importance role of MTA2 in normal development and varieties of human cancers. Except being a component of the NuRD complex with intrinsic histone deacetylase, MTA2 also cooperates with other coregulators in modifying chromatin state to regulate different target genes. We mainly summarize the evidence supporting the view that MTA2 is one of the most frequently genes amplified than mutated in different types of cancer. Clinical studies show the higher expression of MTA2 is usually associated with advanced tumors and might be used as a poor predictor in the prognosis of patients, but the mechanism by which it promotes cancer progression is distinguished with these cancers. Correlating cellular results with mouse conclusion reveal that MTA2 may be an elusive direct target for cancer therapy. Controversy, El-Nikhely et al. (2020) found that MTA2 negatively regulates NF-κB through forming the MTA2/NuRD corepressor complex and interacting with RelA to inhibit lung cancer growth and inflammation. So, it is emergency to take findings from experimental model systems into clinical trial and to confirm whether targeting MTA2 is clinically effective.

Additionally, regarding MTA2, there are still many challenges that need further research. Although multiple articles correlate MTA2 overexpression with poor prognosis, why this MTA2 hasn’t progressed to diagnostic applications. Potential limitations like tumor heterogeneity in biopsy samples or lack of standardized detection protocols remain unknown. Establish MTA2 activity signatures correlated with chromatin accessibility profiles. Validate circulating nucleosome positioning patterns reflecting MTA2-regulated transcriptional footprints might as liquid biopsy markers for early intervention. Moreover, there have been no reported studies on human-targeted therapy for MTA2; only animal-related research exists.

The role of MTA2 in immune regulation exhibits conflicting findings across studies, reflecting its context-dependent interactions within the tumor microenvironment (TME). MTA2 is positively correlated with most immune cells in pan-cancer (Huang et al., 2023). Huang et al. (2023) found that MTA2 was positively associated with immune checkpoint genes, such as CTLA-4, HAVCR2, PD-1, and PD-L1 were highly expressed in the high-MTA2 group, except PD-L2. However, previous reports also discovered that loss of MTA2 function may impair B-cell development and lead to immune system defects (Lu et al., 2019). Moreover, inactivation of MTA2 leads to abnormal T-cell activation and lupus-like autoimmune disease in mice (Lu et al., 2008). The tumor microenvironment has attracted great research and clinical interest as a therapeutic target for cancer (Xiao & Yu, 2021). The immune cells within the tumor microenvironment regulate cancer development (Liu et al., 2024). The tumor-infiltrating T regulatory (Treg) cells are a major obstacle in the cross-talk between CD4 + T cells and CD8 + T cells since they are capable of inhibiting anti-tumour immunity (McRitchie & Akkaya, 2022). High MTA2 expression in LIHC was positively correlated with TME scores in LIHC. It was demonstrated that MTA2 can interact with HDAC2/CHD4 to affect the TME and thus facilitate LIHC formation and progression (Zhang et al., 2021).

This critical reframing would transform the manuscript from a static literature inventory into a dynamic framework for guiding future mechanistic and translational research.

Future gene therapy strategies targeting MTA2 could involve CRISPR-based knockout to suppress its oncogenic activity or siRNA/shRNA delivery to silence its expression. Additionally, engineered transcriptional repressors or small-molecule inhibitors might disrupt MTA2’s interaction with chromatin-remodeling complexes, impairing tumor invasiveness. Combining MTA2-targeted therapies with immune checkpoint inhibitors or CAR-T cells could enhance antitumor efficacy by reversing immunosuppressive microenvironments. Advances in AI-driven protein design or CRISPR prime editing may refine precision, while preclinical models could validate therapeutic potential, paving the way for clinical translation.

In the future, to further validate the feasibility of MTA2 as a target for cancer therapy. We need to employ covalent ligand screening to identify druggable cysteine residues in MTA2’s SANT domain that could enable allosteric disruption of HDAC1 interactions.

Validate MTA2 synthetic lethality partners using CRISPR-Cas12a multiplex screens across cancer lineages with 3D chromatin architecture defects. Profile MTA2-dependent DNA methylation barriers limiting neoantigen presentation, particularly at endogenous retroviral elements in mismatch repair-proficient tumors. Engineer chimeric antigen receptor macrophages (CAR-M) with MTA2-modulated phagocytic checkpoints to overcome CD47-mediated immune evasion. Furthermore, apply single cell multiome sequencing to track MTA2 expression clonal dynamics during metastatic relapse, focusing on its role in maintaining epigenetic plasticity reservoirs. Additional, patient-derived xenograft models are a good way to validate the effects of drug (Li et al., 2018). Moreover, the upstream of MTA2 also provided potential therapy targets, such as sp1 and ETS (Xia & Zhang, 2001).

The authors want to thank BioRender.com for providing the graphic platform.

Additional Information and Declarations

Competing Interests

Author Contributions

Data Availability

The authors declare there are no competing interests.

Xujun Liu conceived and designed the experiments, performed the experiments, prepared figures and/or tables, and approved the final draft.

Yaping Jiang conceived and designed the experiments, authored or reviewed drafts of the article, and approved the final draft.

Yanfeng Hou analyzed the data, authored or reviewed drafts of the article, and approved the final draft.

Xiaoning Li performed the experiments, authored or reviewed drafts of the article, and approved the final draft.

Haixia Li conceived and designed the experiments, authored or reviewed drafts of the article, and approved the final draft.

Wenzhe Si analyzed the data, prepared figures and/or tables, authored or reviewed drafts of the article, and approved the final draft.

The following information was supplied regarding data availability:

This is a literature review.

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
