# Peer review of "The expression of metastasis associated protein 2 in normal development and cancers: mechanism and clinical significance"

_PeerJ, doi:10.7717/peerj.20107_

## Round 0.1 · original submission · Major Revisions

The reviewers found your manuscript interesting and thought it covers an important topic. However, they had several significant concerns that need to be addressed. First, the inclusion and exclusion criteria you used to select the studies you discussed need to be presented in detail. Your manuscript needs to critically evaluate and synthesize the findings of the studies you selected, rather than merely summarizing them. Your discussion of the different studies should be more balanced, rather than overemphasizing certain studies, and any conclusions you make should not be overstated. All the statements or conclusions need to be supported by evidence and to include the appropriate citations. Similarly, the section on MTA2 as a therapeutic target should include relevant studies and any current preclinical or clinical trials. Additionally, any gaps in current knowledge in the field need to be discussed as well as future directions. Lastly, the manuscript needs to be organized so that it has a more logical progression.

Please, submit a detailed rebuttal which shows where and how you have taken all comments and suggestions into consideration. If you do not agree with some of the reviewers’ comments or suggestions, please explain why. Your rebuttal will be critical in making a final decision on your manuscript. Please, note also that your revised version may enter a new round of review by the same or by different reviewers. Therefore, I cannot guarantee that your manuscript will eventually be accepted.

**Language Note:** The review process has identified that the English language must be improved. PeerJ can provide language editing services - please contact us at [email protected] for pricing (be sure to provide your manuscript number and title). Alternatively, you should make your own arrangements to improve the language quality and provide details in your response letter. – PeerJ Staff

Reviewer 1 ·

Basic reporting

The literature references are comprehensive and provide sufficient background and context for the topic. The structure of the article is professional, with well-organized figures and tables that enhance the understanding of the content. The raw data, if applicable, should be shared to ensure transparency and reproducibility.

Experimental design

the rarticle develops a well-supported argument that aligns with the goals set out in the Introduction. It thoroughly discusses the role of MTA2 in various cancers and its clinical significance.

Validity of the findings

The findings are valid and well-supported by the literature. The conclusions are clearly stated and linked to the original research question. The review identifies unresolved questions and future directions, which adds value to the field.

Additional comments

Overall, this is a well-written and comprehensive review that provides valuable insights into the role of MTA2 in normal development and cancer. The authors have done an excellent job of synthesizing a large body of literature and presenting it in a clear and organized manner. The inclusion of figures and tables enhances the understanding of the complex mechanisms discussed. The review is likely to be of significant interest to researchers in the field and could serve as a useful reference for future studies.

Reviewer 2 ·

Basic reporting

First of all, I like the paper. The manuscript presents a comprehensive review of the expression, function, and clinical significance of Metastasis-Associated Protein 2 (MTA2) in normal development and various cancers. It discusses MTA2’s role as a transcriptional regulator affecting key biological processes, including proliferation, apoptosis, DNA repair, and immune cell differentiation. The review also highlights the potential oncogenic role of MTA2 in multiple cancer types, its interactions with transcription factors, microRNAs, and lncRNAs, and its influence on tumor progression. While the manuscript is informative and well-structured, it requires improvements in language clarity, grammatical corrections, and more consistent citation formatting. Additional discussion on potential therapeutic implications and future research directions would further enhance the review.
Comments:
Abstract:
Line 4: "Despite extensive and in-depth research has been constructed, the physiological and pathogenesis of MTA2 is far from being fully understood."Rephrase to: "Despite extensive research, the physiological role and pathogenic mechanisms of MTA2 remain poorly understood."
Line 7: "Accumulated studies have revealed that MTA2 is frequently amplify in several types of cancers..."Rephrase to: "Accumulating evidence suggests that MTA2 is frequently amplified in several types of cancer..."
Line 12: "Substantial evidence indicates the function of MTA2 is depending on the modulation of downstream targets..."Rephrase to: "Substantial evidence indicates that MTA2 functions by modulating downstream targets..."
Introduction:
Line 6: "Identification and characterization of critical genes responsible for tumor progression and metastasis is the focus of numerous investigations all over the world."Update the statistics on overall cancer incidence and the prevalence of this specific cancer type, including survival rates, to emphasize the urgent need for cancer studies. Cite Cancer Statistics, 2024. Additionally, provide a general overview of cancer therapy, referencing the NIH paper“Cancer treatments: Past, present, and future, 2024” for further insights.
Line 20: "Metastasis-associated proteins (MTAs) is such a group of transcriptional co-regulators..."Rephrase to: "Metastasis-associated proteins (MTAs) are a group of transcriptional co-regulators..."
Line 34: "Although both MTA1 and MTA2 are the most up-regulated genes in human cancers, the function of the two proteins are not always overlapping."Rephrase to: "Although both MTA1 and MTA2 are among the most upregulated genes in human cancers, their functions do not always overlap."

Materials & Methods:
Line 10: "Survey methodology"—The section lacks detail on the inclusion and exclusion criteria for literature selection. Consider specifying how studies were chosen.
Results:
Line 17: "MTA2 interacts with multiple signaling pathways that contribute to oncogenesis."Provide references to support this claim.
Line 45: "Studies have demonstrated a correlation between MTA2 overexpression and poor prognosis in gastric, breast, and pancreatic cancers." Cite specific studies that establish this correlation. I think you have to refer to previous TCGA biomarker studies and discuss. So far, there are too many TCGA studies. You should emphasized the TCGA biomarker studies, the following paper should be discussed as examples of prognostic correlation, such as(“Is the voltage-gated sodium channel β3 subunit (SCN3B) a biomarker for glioma?, 2024”,“A Comprehensive Bioinformatic Analysis of Cyclin-dependent Kinase 2 (CDK2) in Glioma, 2022,Clinical powers of Aminoacyl tRNA Synthetase Complex Interacting Multifunctional Protein 1 (AIMP1) for head-neck squamous cell carcinoma, 2022,Potential roles of Cornichon Family AMPA Receptor Auxiliary Protein 4 (CNIH4) in head and neck squamous cell carcinoma, 2022,RAD50 is a potential biomarker for breast cancer diagnosis and prognosis, 2024”)
Discussion:
Line 9: "MTA2 has been shown to interact with transcription factors such as Sp1 and ETS elements in various cancers."Consider expanding on the functional consequences of these interactions.Manypaper used TCGA data to explore MTA2 genes in cancers, Discuss the bias from TCGA, refer to “Genetic expression in cancer research: Challenges and complexity, 2024” and “Technical and Biological Biases in Bulk Transcriptomic Data Mining for Cancer Research, 2025”
Line 27: "The role of MTA2 in immune regulation remains controversial."Provide additional discussion on conflicting findings in the literature. Discuss also the microenvironment of cancer cells, how they can affect immune cells, recent studies in breast cancer microenvironment should be mentioned, such as “Identification of the novel exhausted T cell CD8 + markers in breast cancer, 2024”
Line 48: "Future studies should aim to clarify the upstream regulatory mechanisms of MTA2."Consider suggesting specific experimental approaches for future research.Suggest future studies that could validate these findings in patient-derived xenograft models. Previous studies using xenograft models of cancer should be mentioned, such as “Comparing volatile and intravenous anesthetics in a mouse model of breast cancer metastasis, 2018”
Conclusion:
Line 2: "MTA2 has been identified as a key regulator in tumor progression and metastasis, but its precise role remains to be fully elucidated."Consider discussing potential therapeutic strategies targeting MTA2.

Experimental design

Overall, the manuscript provides an extensive review of MTA2 in cancer biology, but addressing the above comments will improve clarity, strengthen scientific arguments, and enhance readability.

Validity of the findings

First of all, I like the paper. The manuscript presents a comprehensive review of the expression, function, and clinical significance of Metastasis-Associated Protein 2 (MTA2) in normal development and various cancers. It discusses MTA2’s role as a transcriptional regulator affecting key biological processes, including proliferation, apoptosis, DNA repair, and immune cell differentiation. The review also highlights the potential oncogenic role of MTA2 in multiple cancer types, its interactions with transcription factors, microRNAs, and lncRNAs, and its influence on tumor progression. While the manuscript is informative and well-structured, it requires improvements in language clarity, grammatical corrections, and more consistent citation formatting. Additional discussion on potential therapeutic implications and future research directions would further enhance the review.

Reviewer 3 ·

Basic reporting

The manuscript presents a comprehensive summary of MTA2, but the quality of English writing is subpar, making it difficult to follow in many places. There are numerous grammatical errors, awkward phrasings, and repetitive statements. Example: “Accumulated studies have revealed that MTA2 is frequently amplify in several types of cancers…” → should be corrected to “Accumulated studies have revealed that MTA2 is frequently amplified in several types of cancers…”

The authors should seek professional English editing to improve readability. Additionally, the structure of the review lacks clear transitions between sections, leading to redundancy and loss of focus.

Although the review cites a broad range of studies, there is no mention of the methodology used to select these sources. The authors do not specify whether this was a systematic review or a narrative review, making it unclear if the selection process was rigorous. Some key studies on MTA2 and metastasis appear to be missing, while others are over-relied upon. A more balanced and structured approach to literature citation is needed.

Experimental design

The manuscript covers various aspects of MTA2’s function in cancer progression but does not critically evaluate the findings. Instead, it merely summarizes existing research without discussing limitations, contradictory results, or gaps in knowledge. A good review should not just list studies but synthesize findings to identify trends and unresolved questions.

Furthermore, the review does not incorporate a clear framework to categorize the data. The discussion jumps between different cancer types and pathways without a logical progression, making it difficult to grasp the overall significance of MTA2 in cancer. A well-defined structure, possibly grouping studies by cancer type or mechanistic function, would enhance clarity.

Validity of the findings

The review provides useful mechanistic insights into MTA2’s role in chromatin remodeling and tumor progression, but the conclusions are often overgeneralized. The authors repeatedly suggest that MTA2 is a key oncogene without addressing counter-evidence showing that its role can be context-dependent. Some studies have indicated that MTA2 might not always promote metastasis, yet these perspectives are missing from the manuscript. The review should include a discussion on cases where MTA2 does not function as an oncogene or where its effects are unclear.

Another issue is the speculation on MTA2 as a direct therapeutic target. The manuscript suggests that targeting MTA2 could be a viable strategy, but there is no mention of clinical trials or preclinical studies testing this hypothesis. Without such evidence, this claim remains speculative and should be toned down or supported with additional references.

Additional comments

The conclusion does not effectively summarize the main points of the review. Instead of simply reiterating MTA2’s involvement in metastasis, it should highlight key research gaps and propose future directions. A well-written conclusion should provide actionable insights, guiding future research rather than restating known information.

---

## Round 0.2 · accepted · Accept

Thank you for thoroughly addressing the reviewers' comments and thus greatly improving your manuscript.

Reviewer 2 ·

Basic reporting

ok

Experimental design

ok

Validity of the findings

ok

Additional comments

ok